# Role of Cytochrome P450 Enzyme in Plant Microorganisms’ Communication: A Focus on Grapevine

**DOI:** 10.3390/ijms24054695

**Published:** 2023-02-28

**Authors:** Daniela Minerdi, Stefania Savoi, Paolo Sabbatini

**Affiliations:** 1Department of Agricultural, Forestry and Food Sciences, University of Turin, Largo Paolo Braccini 2, 10095 Grugliasco, TO, Italy; 2Department of Horticulture, Michigan State University, East Lansing, MI 48824, USA

**Keywords:** plants, grapevine, cytochrome P450 enzyme, cypome, interactions between plants and microorganisms, *Vitis vinifera*

## Abstract

Cytochromes P450 are ancient enzymes diffused in organisms belonging to all kingdoms of life, including viruses, with the largest number of P450 genes found in plants. The functional characterization of cytochromes P450 has been extensively investigated in mammals, where these enzymes are involved in the metabolism of drugs and in the detoxification of pollutants and toxic chemicals. The aim of this work is to present an overview of the often disregarded role of the cytochrome P450 enzymes in mediating the interaction between plants and microorganisms. Quite recently, several research groups have started to investigate the role of P450 enzymes in the interactions between plants and (micro)organisms, focusing on the holobiont *Vitis vinifera.* Grapevines live in close association with large numbers of microorganisms and interact with each other, regulating several vine physiological functions, from biotic and abiotic stress tolerance to fruit quality at harvest.

## 1. Cytochromes P450 Enzymes: General Features

Cytochromes P450 (P450s or CYPs) are heme–thiolate terminal monooxygenases that transfer one atom from molecular oxygen to X–H bonds, where X may be the -C, -N, or S of a substrate, with a concomitant reduction in the remaining oxygen atom to water [1]. The CYP-mediated molecular oxygen activation gives rise to ketones, alcohols, aldehydes, epoxides, and carboxylic acids. Radicals or reactive oxygenated intermediates generated by CYPs can also lead to the formation of oxidative C–C bond cleavage [2], dealkylation [3], desaturation [4], dehydration [5], the oxidative rearrangement of carbon skeletons [6], and decarboxylation [7]. CYPs are a large superfamily of enzymes that show an absorbance peak at 450 nm when their heme is reduced and complexed with carbon monoxide. These enzymes were discovered in 1958, while studying the spectrophotometric properties of pigments in a microsomal fraction of rat livers [8]. A few years later, it was demonstrated that this pigment is a heme-containing protein, and it was called “P-450” (P for pigment) [9]. The heme group plays essential roles in catalysis, providing the P450s’ carbon monoxide-binding ability; moreover, the signature sequence FxxGxxxCxG in the heme-binding domain functions as the fifth ligand to the heme iron. CYPs are ancient enzymes that are extensively diffused in the organisms of all kingdoms of life (Figure 1), including viruses [10], with the largest number of P450 genes being present in plants [11]. Plant P450s are classified into the A-type group and are involved in plant-specific biochemical pathways [12]. The non-A-type P450 enzymes [13] form distinct clades and are more closely related to non-plant P450 enzymes than to the A-type. CYPs are classified into families on the basis of the sequence identity of their amino acid sequences, and the symbol *CYP* is the root followed by an Arabic numeral representing family therefore, the subfamily is indicated by a letter and the gene is represented by a number [14]. The features and functions of P450 enzymes have been studied from bacteria to mammals (see http://drnelson.uthsc.edu/CytochromeP450.html, accessed on 15 November 2022) [14]. Human P450 enzymes metabolize drugs and synthesize endogenous compounds essential for human physiology. Often, alterations in specific P450s affect the biological processes that they mediate, leading to a disease [15].

## 2. Cytochrome P450 Reactivity

CYPs, through the activation of molecular oxygen, are able to monooxygenate the substrate, followed by the insertion of a single atom of oxygen into an organic substrate and a reduction in the another oxygen atom to water according to the following scheme:RH + O_2_ + NAD(P)H + H^+^ → ROH + H_2_O + NAD(P)^+^

NAD(P)H supplies electron equivalents through an electron transfer chain that involves different redox partners, the nature of which is utilized to classify cytochromes P450 into 10 classes [16]. The organic substrate (R) binds to the heme group of the enzyme and this binding allows the transfer of an electron from NADPH through cytochrome P450 reductase (CPR) to the hem domain that reduces the iron (Fe) from the ferric state (Fe^3+^) to the ferrous state (Fe^2+^). The molecular oxygen binds to ferrous P450, forming a ferrous CYP–dioxygen complex, and the second electron is transferred from CPR to the ferrous CYP–dioxygen complex, forming the peroxo complex, and this complex is rapidly protonated twice to form one molecule of water and an iron–oxo complex. The oxidized reaction product (RO) is formed during the last step of the catalytic cycle when the atom of oxygen in the iron–oxo complex binds to the organic substrate (R) [17] (Figure 2).

## 3. The Role of Cytochromes P450 in Plants 

In plant cytochromes, P450 enzymes are present in several organelles and organs, namely, the endoplasmic reticulum, mitochondria, and chloroplast [18]. They are also found in shoot, bulbs, hypocotyl, roots, and the endosperm of germinating seeds. In both plants and animals, CYPs have an absorbance range from 447 to 452 nm and constitute the largest family of enzymes in the metabolism of a plant, where they are grouped into ten clans [19,20]. The most studied family is CYP51, with a pivotal role in regulating the synthesis of sterols [21] and triterpenes [22]. CYP71 impacts shikimate products and intermediates [23,24]. The CYP72 clan has a role in the catabolism of isoprenoid hormones [25], and the CYP74 synthetizes oxylipin derivatives and allene oxide in the octadecanoid and jasmonate pathways [26,27]. CYP85 participates in the metabolism of cyclic terpenes and sterols in the brassinosteroid and in abscisic acid and gibberellin pathways [28]. The CYP86 hydroxylates fatty acids [29], the CYP97 hydroxylates carotenoids [30], and CYP710 control the synthesis of sterol C-22 desaturase [5]. The CYP450 family in plants has a molecular mass between 45 and 62 kDa [4,31,32]. In plants, P450 enzymes are involved in the biosynthesis of structural polymers, defense against pathogen infection, communication with other organisms, hormonal signaling, herbicide resistance, and stress tolerance (Figure 3). 

Plant hormone metabolism is also regulated by cytochrome P450 enzymes that control cell division and cell expansion, vascular differentiation, fruit growth, root development, and flower formation. Cytochromes P450 protect plants from dehydration [3], UV stress [33], and drought [34]. Moreover, P450s play a role in physiological process, such as detoxification, adaptation responses to heavy metals, salts, and chemicals (e.g., herbicides) [35,36]. In addition, many P450s participate in the biosynthesis of cell wall components [37]. CYP97 engages in the biosynthetic pathway of xanthophylls, hydroxylating the beta and gamma rings of carotenoids [38] (Table 1).

In *Solanum lycopersicum*, the CYP78 subfamily modulates the number and length of the lateral shoots and fruit ripening time [39]. In *Solanum tuberosum*, CYP72 is involved in the biosynthesis of two steroidal glycoalkaloids that catalyze the 26- and 22-hydroxylation steps. The knockdown plants of CYP72 are sterile, and tubers do not sprout during storage. 

Tuber sprouting and the biosynthesis of glycoalkaloid in potatoes, two traits that significantly impact potato breeding and are important for the industry, can be controlled by tools provided by the functional analyses of the CYP72 family [40]. The cuticle of anther and the pollen exine are protective envelopes of the male gametophyte and the pollen grain. The fatty acid hydroxylase CYP703 is essential for male fertility in rice because the CYP703A3-2 mutant shows that pollen exine is defective and that the anthers exhibit a reduced level of cutin monomers and wax components [41]. Cucumber and melon contain cucurbitacins, highly oxygenated triterpenoids responsible for the bitter taste [47]. Cucurbitacins derive from the tetracyclic triterpenoid cucurbitadienol that is further oxidized to cucurbitacin at different positions by CYP81 and CYP87 [42,43,48] and CYP72 and CYP71 [22,44]. In *Olea europaea*, CYP72 catalyzes the oxidative C–C bond cleavage in the biosynthesis of secoxy-iridoids, a series of phenolic compounds that are essential for the flavor and quality of olive oil [45]. CYP716 is involved in the biosynthesis of the triterpenic acids present in apple fruit [46], and CYP97 is involved in the biosynthetic pathway of lutein in *Sorghum* [32].

## 4. Cytochrome P450 Enzyme in Plant–Microorganism Interaction 

The main focus of this review is an overview of the role of the cytochrome P450 enzyme in plants, including crop plants, focusing on their involvement in the interaction between plants and (micro)organisms (Figure 4). In this context, biotic stresses were considered as a consequence of the plant interaction with the organism, causing the stress (e.g., pathogens). CYPs contribute to the synthesis of terpenoids, phytoalexins, alkaloids, and cyanogenic glucosides, which play important roles in the response of plant to biotic stresses.

### 4.1. CYPs in Plant–Bacteria Interaction 

In *Coptis japonica*, the biosynthesis of berberine, an antimicrobial benzylisoquinoline, is mediated by the methylenedioxy bridge-forming enzyme CYP719 [49]. In potatoes, the antimicrobial compound oxylipin is synthetized by the pathogen-inducible divinyl ether synthase CYP74 [50]. When the bacterial pathogen *Pseudomonas syringae* infects *A. thaliana*, it causes a hypersensitive response that induces the expression of the *CYP76C2* gene [51,52]. *CYP76C2* gene expression is associated with leaf senescence, the aging of cell cultures, and wounding [51,53]. The *pepCYP* gene product plays a role in the defense mechanism when the pathogenic fungus *Colletotrichum gloeosporioides* invades and colonizes the fruits of pepper [54]. CYP89 is another cytochrome P450 in pepper that plays a pivotal role in plant defense response against pathogen infections [55]. Isoflavonoids are phenylpropanoid metabolites that act as antimicrobial compounds [56] that are present in legumes [57]. CYP81 and CYP93 are involved in isoflavonoid biosynthesis in *Medicago truncatula* [58], *Glycyrrhiza echinata* [59], and *Cicer arietnum* [60]. In response to bacterial and fungal pathogen attack, many plants rapidly induce the biosynthesis of phytoalexins, which are low molecular weight antimicrobial compounds showing great structural diversity. CYP71 and CYP79 are responsible for the biosynthesis of the phytalexin camalexin in *A. thaliana* [13].

### 4.2. Cytochrome P450 Enzyme in Plant–Fungi Interaction 

In fungi, cytochrome P450 enzymes regulate a plethora of different physiological mechanisms including fertility and fitness. In filamentous fungi, P450 enzymes play a housekeeping role, being involved in sterol synthesis [61], and CYP51 has been identified as a good target for the development of antifungal drugs [62]. The diversification and expansion of the P450 families has been related to fungal pathogenicity [63,64]. P450 enzymes are involved in the responses of the host plants to pathogen attacks, neutralizing the production of antibiotic phytoalexins [65]. CYP57 is a pisatin demethylase of the fungus *Nectria haematococca*, a pathoghen of peas. This P450 enzyme inactivates the isoflavonoid pisatin that is produced by peas when the fungus invades the plant [16]. In *Nicotiana benthamiana*, CYP51 participates in the synthesis of antimicrobial triterpenes and is involved in antifungal activity [22]. In cotton, CYP70 participates in the synthesis of gossypol and related sesquiterpene induced by *Verticillium dahlia* infection [66]. The fungus *Fusarium oxysporum* causes severe vascular wilt disease in several crop plants [67]. Antagonistic *F. oxysporum* strains protect plants from pathogenic fungi, and they have been successfully used as biological control agents in agriculture [68]. The CYP505 family belonging to class VIII [16] includes a flavocytochrome wherein an N-terminal heme domain is fused to the C terminal FAD/FMN reductase domain [69]. CYP505 members hydroxylate fatty acids in the subterminal omega position, a step that is fundamental for these molecules to be used as an energy source [70]. Oxidized fatty acids are endogenous signal molecules with an important role in the activation of plant defense mechanisms [71] during interactions between plants and fungi [72]. Pathogenic *formae specialis* of *F. oxysporum* have a CYP505 enzyme that is differentially expressed when the fungus behaves as a pathogen or as an antagonist. In a particular case, using lettuce plants, it was found that *F. oxysporum CYP505A1* is expressed in the host plant during the early phases of the interaction, both in pathogenesis and in antagonism, while its expression is silenced only in the late phase when the fungus behaves as a pathogen [73].

CYP505A1 can mono-hydroxylate lauric, palmitic, and stearic acids present in the cortical cell membranes of the plant, and these hydroxylated compounds might activate the plant defense system [73]. There is increasing evidence that P450 enzymes have a pivotal role in plant defense from pathogenic fungi, and the mechanism can be summarized as the activation of the synthesis of secondary metabolites [74]. Other examples of fungal P450 upregulation are found during the invasion of *Heterobasidion annosums* [75], *Moniliophthora perniciosa* [64], and *Botrytis cinerea* [76], specifically of their host plants. *F. oxysporum* f. sp. *cubense*, which is pathogenic to banana, upregulates the expression of its nitric oxide reductase CYP55 when it invades the plant [77]. CYP55 is involved in the nitrogen response pathway that is essential for fungal pathogenicity [78]. CYP55 is expressed in *F. oxysporum* f.sp. *vasinfectum* during the invasion of the roots of the cotton plant [78]. A cytochrome P450 that is involved in the metabolism of sulfacetamide, a secondary metabolite that is important for fungal pathogenesis, is upregulated in *Verticillium dahlia* during the first days of infection in cotton [79]. *Fusarium graminearum* induces the expression of a benzoate 4-monooxygenase cytochrome P450 gene when it invades wheat coleoptiles [79].

Arbuscular mycorrhizas (AMs) are symbiotic associations between the roots of land plants and fungi belonging the phylum Glomeromycota [80]. The fungal hyphopodium invades the root tissue forming the arbuscule, the site where nutrients between the plant and the fungus are exchanged. It has been shown that, when the AM symbiosis is forming between *Lotus japonicus* and *Rhizopus irregularis*, there is a high level of expression of *cyp* genes in the intraradical hyphae, which due to their involvment in the synthesis of sterols for membrane biogenesis during arbuscule formation [81].

## 5. Cytochrome P450 Enzyme in *Vitis vinifera*

This section is focused on *Vitis vinifera* and the role of cytochromes P450 in grapevine metabolism and during the interaction between grapevines and (micro) organisms. In this context, the interaction of grapevines both above and below ground with several organisms, such as arthropods (spiders, mites, insects) and microorganisms (fungi, oomycetes, bacteria, viruses), nematodes, birds, and mammals, as well as other plants is pivotal in understanding the complex productive system of a vineyard.

The grapevine is a perennial plant of extremely high economic importance worldwide. In vineyards, it lives in association with many microorganisms. Grapevines, together with several fruit crops, are highly valuable but most of the premium cultivars used for winemaking, especially the extensively used European *Vitis vinifera* cultivars, are highly susceptible to several pathogenic microorganisms including fungi, bacteria, phytoplasma, and viruses. In fact, grapevines interact both above and below ground with arthropods (e.g., insects) microorganisms (e.g., fungi, bacteria, viruses), nematodes, animals (e.g., birds and mammals), and other plants. Many interactions with different organisms are beneficial, providing ecological “services” to the vineyards. The classical example is related to earthworms, which improves soil fertility by transforming organic matter into humus, creating soil porosity where the roots can penetrate and actively up- take nutrients in symbiosis with soil microorganisms. Bacteria, fungi, and viruses are organized in communities and everyone can be beneficial, neutral, or harmful to the grapevines. In this context, microorganisms interact with each other and impact several plant functions. Therefore, the grapevine associated with its microbial communities constitutes a supra-organism, called holobiont, and the mechanism of functioning is related to plant–microorganism interactions. In this sense, even the overall health of the grapevine is influenced by the diversity and structure of the microbial communities. The microbiome is a pivotal component of the grapevine, and it is also extremely responsive to biotic and abiotic changes [82]. The close interaction of the grapevines with many microorganisms modulates vine physiology throughout the entire growing season and during all the phenological stages, impacting grapevine holobiont health and above all the fruit chemical makeup at harvest, which consequently affects wine quality. The omics sciences have delivered a new era in plant biology, starting with the precise description of the taxonomic composition of the microbial communities of the grapevines in each single organ and during their annual growth and development. Following those discoveries, scientists are approaching the complex understanding of grapevine microorganisms and the specific functional contributions of microbes to the grapevine holobiont. To date, few studies have addressed the functional characteristics of the microbiota through metabolomic and transcriptomic approaches in the grapevine holobiont. However, the understanding of the complexity of the grapevine holobiont is a pivotal issue for the future of the wine industry worldwide. The potential understanding and consequent manipulation of the microbiota in a grapevine holobiont can lead to better vineyard management focused on urgent reduction of pesticide and chemical fertilizer inputs through biocontrol, which are key factors for the development of the sustainable viticulture of the future.

### 5.1. Role of Cytochrome P450 in Vitis vinifera 

*Vitis vinifera* has 579 cytochrome P450 sequences belonging to 48 families, a number that is similar to their number in *A. thaliana* (242), *S. lycopersicum* (272), and *Oryza sativa* (309) [83]. Most of the CYP sequences in the grapevine genome are organized in clusters originating from tandem or segmental duplications. Some grapewine P450 genes are induced upon biotic stress, while others are specifically activated during grape berry ripening and might have a role in the production of specific volatiles in berries such as aroma compounds (Figure 5). *CYP71*, *CYP72*, *CYP75*, *CYP76*, *CYP81*, *CYP82*, and *CYP89* genes are enlarged in the grapevine genome and are involved in secondary metabolism and are differentially expressed during four different stages of berry development [83,84,85].

The aroma of wine consists of a hundred different volatile compounds at concentrations spanning several orders of magnitude [86]. A characteristic wine aroma depends on trace components with very strong odors [86]. Among monoterpenes, which are important constituents of the aromas in wines, wine lactone is a bicyclic monoterpene lactone that has the most potent odor [87]. (*E*)-8-carboxylinalool is present both in grapes and wines as a glucose ester with a sugar moiety attached to the carboxyl functional group [88]. *CYP76* is highly expressed in maturing *V. vinifera* berries and oxidizes linalool to (*E*)-8-carboxylinalool, which acts as a precursor to wine lactone with a nonenzymatic step during wine maturation and aging [83] (Figure 4).

### 5.2. Cytochromes P450 in Vitis vinifera Interaction with Microorganisms

*V. vinifera* is highly susceptible to many pests and pathogens, causing great economic loss each year. Currently, we are at the beginning of discovering the involvement of cytochrome P450 enzymes in the interaction between grapevine and its pathogens. Some of the most dangerous pathogens include the bacterium *Xylella fastidiosa*, the phytopathogenic bacterium *Candidatus*, phytoplasma *solani*, and the phytoplasma *Flavescence dorée.* The overexpression of plant gene coding for cytochrome P450 enzymes after the invasion of the pathogen has been proved for the three microorganisms mentioned above. In detail, CYP736 was involved in the host response to *X. fastidiosa* infection [89]; moreover, the gene coding for a cytochrome P450 enzyme belonging to CYP2C family was found to be overexpressed after the infection of *Candidatus* phytoplasma *solani* [90], and the invasion of *Flavescence dorée* induced the over-expression of CYP86 [91] (Figure 4).

## 6. Future Directions and Concluding Remarks

The plant microbiome is a key determinant of plant health and productivity, as well as all tissues of a plant host’s microbial community [92]. The manipulation of the plant microbiome has the potential to reduce the incidence of plant disease [93], increase agricultural production [94], reduce chemical inputs [95], and reduce emissions of greenhouse gasses [96], resulting in more sustainable agriculture. Due to the importance of microbiomes and the cypome complement of both plants and microbes, it is important to decipher the cypome composition of the plant microbiomes and its contribution to the metabolism of the plant as well as to understand the influence of the plant on the expression of the cypome of its microbiome. In recent years, our understanding of the molecular aspects of the grapevine microbiome has greatly increased. However, a lot of research remains to be carried out to precisely decipher and finely characterize the different aspects of microorganisms’detection by the grapevines and the subsequent activation and establishment of physiological plant reactions. This applies, for example, to the precise characterization of the role of P450 in comparative studies of the genetic diversity of resistance genes and other defense-related genes in various cultivated and wild grapevines. The enormous impact of the obtained knowledge from grapevine interaction with the microbiome has already contributed, as evidenced from its integration in breeding experiments, either through genetic transformation or through marker-assisted selection, and this research approach will surely become more common in the future.

## Figures and Tables

**Figure 1 ijms-24-04695-f001:**
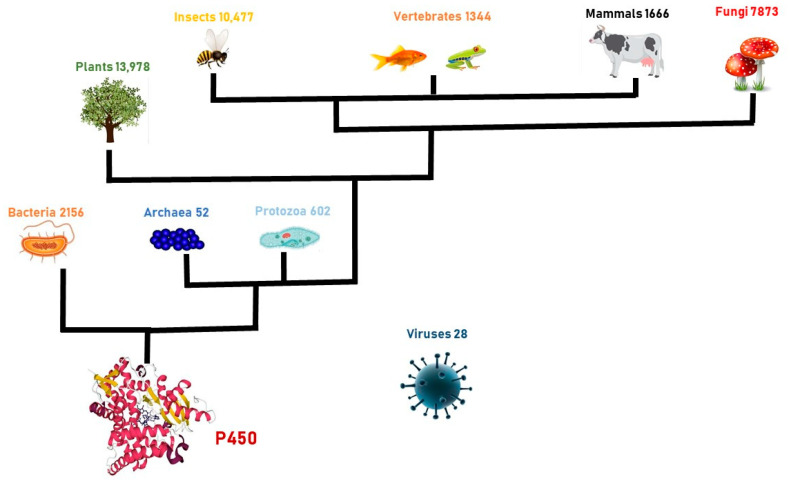
Diffusion of cytochrome P450 enzyme in the kingdom of Life.

**Figure 2 ijms-24-04695-f002:**
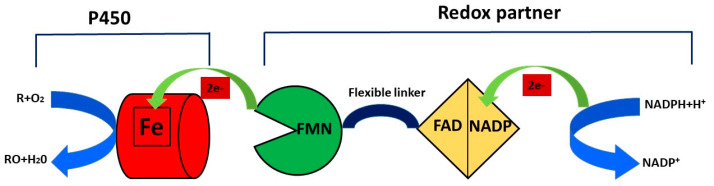
Catalytic cycle of cytochrome P450 enzyme.

**Figure 3 ijms-24-04695-f003:**
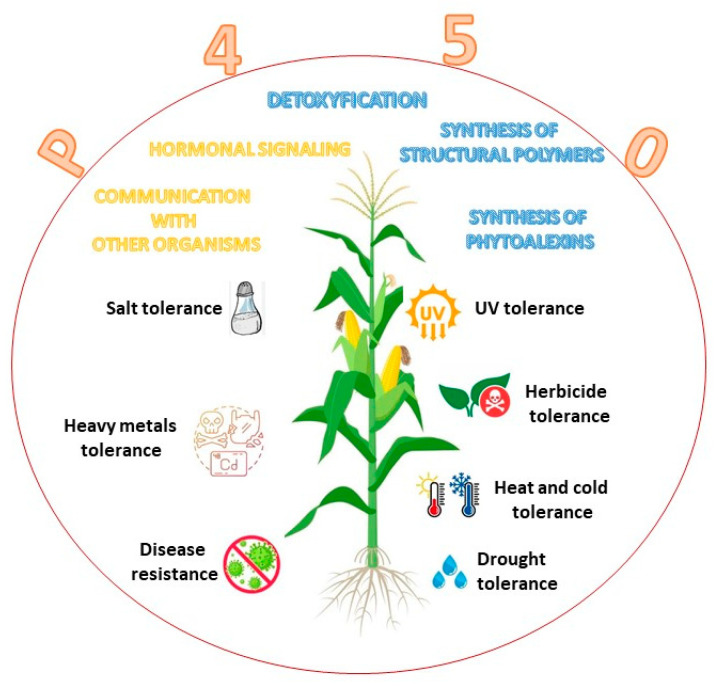
Role of cytochrome P450 enzyme in plants.

**Figure 4 ijms-24-04695-f004:**
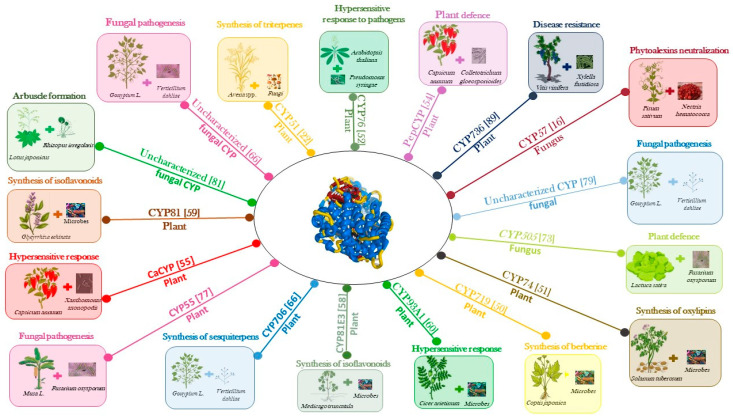
Role of cytochrome P450 enzyme in plant–microorganism interactions.

**Figure 5 ijms-24-04695-f005:**
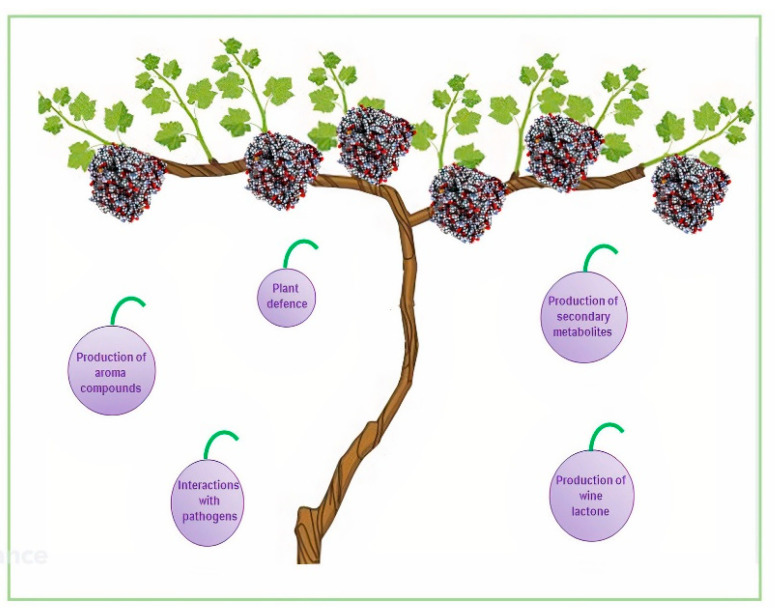
Role of cytochrome P450 enzyme in *Vitis vinifera*.

**Table 1 ijms-24-04695-t001:** Role of cytochrome P450 enzyme in crop plants.

Plant	P450	Function	Reference
*Solanum lycopersicum*	CYP789A	Fruit size Plant architecture	[39]
*Solanum tuberosum*	CYP72A208 CYP188	Glycolakaloid biosynthesis	[40]
*Oryza sativa*	CYP703A3	Male fertility	[41]
*Cucumis sativus*	CYP88L2 CYP81Q58 CYP78D20	Synthesis of cucurbitacin	[42,43]
*Maesa lanceolata*	CYP72A CYP76A	Synthesis of triterpenes	[22,44]
*Olea europea*	CYP72	Synthesis of secoxy-iridoids	[45]
*Malus domestica*	CYP716A CYP175	Synthesis of triterpenic acids	[46]
*Sorghum bicolor*	CYP97C1 CYPP97A3	Synthesis of lutein	[32]

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
