# Peer review of "Role of Cytochrome P450 Enzyme in Plant Microorganisms’ Communication: A Focus on Grapevine"

_ijms, 2023, doi:10.3390/ijms24054695_

Round 1
Reviewer 1 Report
The authors reviewed the role of P450 in crops in details. The review is extensive. However, it requires a bit more work on the logic of section 1: general features. There seems a ton of same contents repeating again and again, such as line 40-42, 55-59 etc. Section 7 don't need to be separate out of section, it could be combined with section 8. Lastly, there are some spelling and grammar errors the authors need to be careful of.
Author Response
The authors reviewed the role of P450 in crops in details. The review is extensive. However, it requires a bit more work on the logic of section 1: general features. There seems a ton of same contents repeating again and again, such as line 40-42, 55-59 etc.
Response: Section 1 in the present version of the manuscript has been revised. Now it is more focused and it has been shortened to avoid repetitions.
Section 7 don't need to be separate out of section, it could be combined with section 8.
Response: Sections 7 and 8 in the new version of the manuscript are combined with section 4, and organized in subsection 4.1 and 4.2 respectively.
Lastly, there are some spelling and grammar errors the authors need to be careful of.
Response: Spelling and grammar has been carefully checked.
New structure:
- Cytochromes P450 enzymes: general feature
1.1 Cytochrome P450 reactivity
- The role of cytochromes P450 in plants
2.1 Cytochrome P450 in crop plants
- Cytochrome P450 enzyme in plant-microorganisms’ interaction
3.1 CYPs in plant-bacteria’s interaction
3.2 Cytochrome P450 enzyme in plant-fungi’s interaction
- Cytochrome P450 enzyme in Vitis vinifera
4.1 Vitis vinifera as a holobiont
4.2 Role of cytochrome P450 in Vitis vinifera
4.3 Cytochromes P450 in Vitis vinifera interaction with microorganism
- Future directions and concluding remarks
Reviewer 2 Report
This mansucript entiteld "Role of cytochrome P450 enzyme in plant microorganisms’ communication: a focus on grapevine" reviewed function of P450 in interaction between plant and microbe, especially grapevine. However, the organization of this manuscript is not well. And this review can not provide valuable points for readers. Authors should improve it. Many comments in attachment files. please check it.

Author Response
This mansucript entiteld "Role of cytochrome P450 enzyme in plant microorganisms’ communication: a focus on grapevine" reviewed function of P450 in interaction between plant and microbe, especially grapevine.
However, the organization of this manuscript is not well.
Response: In the new version of the manuscript the organization has been changed and improved (see coments above to the REV 1)
And this review can not provide valuable points for readers.
Response: We strongly believe, as recognized by the other reviewers, that we have several valuable points for the readers. The new revised version of the manuscript was improved to make the concepts more clear. In particular, with the support of the several positive comments of the other reviewers, we are convinced that this review presents a novel topic providing readers with a new line of thought focused on the interaction between grapevines and microorganisms mediated by the cytochrome P450. In addition, our review propose a novel approach in considering the cytochrome P450 enzyme gene that coordinate the interaction between grapevines and microorganisms in future plant breeding approaches.
Authors should improve it.
Response: The reviewer missed to specify what to improve. However, the present version of the manuscript has been strongly improved, starting from the changing of the overall organization of the review.
Many comments in attachment files. please check it.
Response: All the comments have been checked. Namely:
Chapter 3 and 4 has been combined: chapter 3 is now become chapter 2 and chapter 4 is now section 2.1. For what concerns the question “I can not find any relationship between Microbiome P450” written by the reviewer, it is true and that is the reason why we stressed in our review to investigate this relationship and it is reported in the “future direction” section of the manuscript.
Reviewer 3 Report
In this paper, cytochrome P450 was introduced from the general characteristics to the catalytic reaction process, and then the important role of P450 in the interaction between plants and microorganisms was discussed. In particular, the interaction between P450 enzyme in grape fruit quality and related microorganisms was introduced. In addition, it is proposed to consider applying the cytochrome P450 enzyme gene that produces interaction between plants and microorganisms to plant breeding in the future. This paper presents a novel topic, detailed content and appropriate references, providing readers with a new direction of thinking, that is, focusing on the interaction between plants and microorganisms with cytochrome P450. However, there are still some writing defects that need to be modified. It is recommended to accept them after modification.
1. The structure of the article is not clear. The author clearly takes grapes as an example in the title. When introducing the contents related to grapes, it is suggested to combine the 7, 8 and 9 chapters into one big chapter, and then divide them into smaller chapters, such as 7,7.1,7.2 and 7.3.
2. Some chapters are repeated. Both chapters 3 and 4 introduce the role of cytochrome P450 in plants and suggest that they be combined. Chapters 5 and chapter 6 overlap or chapter 5 contains chapter 6. It is suggested to merge or divide into chapters, such as 5, 5.1 and 5.2; Chapter 8 and 9 are about grape P450 enzyme and its function, it is suggested to combine;
3. There is only one sentence in the second paragraph of the chapter 4, which is suggested to be merged with the previous content;
4. It is suggested to carefully review the format of the reference to make it meet the requirements of the journal. For example, reference 71 in line 476 does not indicate the page number.
Author Response
In this paper, cytochrome P450 was introduced from the general characteristics to the catalytic reaction process, and then the important role of P450 in the interaction between plants and microorganisms was discussed. In particular, the interaction between P450 enzyme in grape fruit quality and related microorganisms was introduced. In addition, it is proposed to consider applying the cytochrome P450 enzyme gene that produces interaction between plants and microorganisms to plant breeding in the future. This paper presents a novel topic, detailed content and appropriate references, providing readers with a new direction of thinking, that is, focusing on the interaction between plants and microorganisms with cytochrome P450. However, there are still some writing defects that need to be modified. It is recommended to accept them after modification.
- The structure of the article is not clear.
Response: The structure has been changed to make it more clear to the readers.
- The author clearly takes grapes as an example in the title. When introducing the contents related to grapes, it is suggested to combine the 7, 8 and 9 chapters into one big chapter, and then divide them into smaller chapters, such as 7,7.1,7.2 and 7.3.
Response: We followed the suggestion of the reviewer and in the new version of the manuscript, the chapters 7, 8 and 9 have been combined in chapter 4 and sub-sections 4.2, 4.2 and 4.3, respectively.
- Some chapters are repeated. Both chapters 3 and 4 introduce the role of cytochrome P450 in plants and suggest that they be combined.
Response: We followed this important suggestion of the reviewer to improve the clarity of our manuscript of the referee and in the revised version chapter 3 and 4 has been combined: chapter 3 is now become chapter 2 and chapter 4 is now section 2.1.
- Chapters 5 and chater 6 overlap or chapter 5 contains chapter 6. It is suggested to merge or divide into chapters, such as 5, 5.1 and 5.2;
Response: Chapters 5 and 6 have been merged into chapter 3 and divided into sub-sections 3,1 and 3.2, respectively.
- Chapter 8 and 9 are about grape P450 enzyme and its function, it is suggested to combine;
Response: Chapters 8 and 9 have been combined into chapter 4.
- There is only one sentence in the second paragraph of the chapter 4, which is suggested to be merged with the previous content
Response: The sentence has been merged with chapter 2.
- It is suggested to carefully review the format of the reference to make it meet the requirements of the journal. For example, reference 71 in line 476 does not indicate the page number.
Response: The format of the references has been carefully checked.
Round 2
Reviewer 3 Report
It is recommended to accept it after modification.
1. Part 1.1 describes the reaction of P450, which is independent of the previous section and can be divided into separate sections, such as the second section.
2. Please check whether the format [23] [24] appears in the reference position of the paper meets the requirements of the magazine.
3. The word "organels" in the first sentence of Part 2 is misspelled; it should be "organells."
4.Formatting errors, situations in which a review of previous articles in the journal failed to find content between headings and subheadings, such as between 3 and 3.1 or 4 and 4.1 in this manuscript.
5. The manuscript still did not make clear the relationship between 2 and 2.1, and either merged or removed 2.1.
Author Response
Dear Editor,
Below is a point-by-point response to your comments and concerns. I hope that the revised version now meets the standard for publication in “International Journal of molecular Sciences”.
Kind regards,
Daniela Minerdi
The manuscript focuses on species interactions between plants and microorganisms, which involve chemical exchanges and gene interactions between species, and these types of topics are all very fascinating. However, the shortcomings of the manuscript are also very evident, particularly the issues raised by reviewer 3.
The paper is not well organized, and the problems remained even after revision. For example, in section 4, branch 4.1 is listed but 4.2 is not visible.
We have made revisions to our manuscript as per Reviewer n. 3's suggestion and removed paragraph 4.2. However, upon further consideration, we believe that your comment is valid, and we have decided to include paragraph 4.2 back into the manuscript as it was in the original version. We fully agree with you that including paragraph 4.2 will significantly improve the clarity of the manuscript for readers of the IJMS.
Thank you for your guidance in this matter.
Sections 4 and 5 should be the main contents of this manuscript. The interactions mentioned in part 4 are also very complex. And here CYP51 and sterol synthesis are included. This is a complex network, so I strongly recommended the authors comb through the literature to give a more visual diagram rather than simply listing information in a table.
Dear Editor, we would like to inform you that we have taken your suggestion into consideration and created a new figure named Figure 4 to provide a clearer representation of the complex interactions and network described in paragraph 5 of our manuscript.
